# DATA DIVERSITY FOR COMPOSITIONAL GENERALIZATION

## ABSTRACT

Human cognition excels at understanding complex concepts by composing simpler learned elements, enabling efficient learning and generalization to novel scenarios. Recent studies suggest that machine learning models may exhibit similar compositional capabilities by first acquiring fundamental components and then recombining them. A key driver of this process is the training data, and in particular, the statistical diversity of that data plays a crucial role in shaping a model's ability to generalize. In this work, we introduce a framework that disentangles the multifaceted notion of diversity and formalize its impact on model performance and generalization ability from different perspectives. Through both theoretical analysis and empirical validation, we demonstrate that simply increasing data diversity without a principled strategy does not necessarily lead to improved generalization. Instead, achieving optimal generalization requires a principled understanding of data diversity. Guided by this insight, we propose a high-level set of guidelines for constructing and curating training datasets that facilitate more efficient learning and better generalization to unseen compositions.

## 1 INTRODUCTION

In recent years, artificial intelligence (AI) has achieved remarkable progress across a broad range of tasks, including language understanding and generation (Hurst et al., 2024; Liu et al., 2024; Grattafiori et al., 2024), reasoning and planning (Guo et al., 2025; Jaech et al., 2024), image and video generation (Dhariwal & Nichol, 2021; Wang et al., 2025), to name only a few. A major driver of this success is the unprecedented scale of training data (Kaplan et al., 2020). Modern AI systems are trained on data volumes that far exceed human experience, by some estimates, more than 200 lifetimes' worth (Griffiths, 2020). As the scale of training data continues to grow and we are approaching practical and economic limits on collecting even more data for training (Villalobos et al., 2022; Muennighoff et al., 2023), improving data efficiency has become increasingly important. While substantial effort has devoted to improving data quality and developing data selection strategies (Xia et al., 2024; Albalak et al., 2024; Xie et al., 2023; Gu et al., 2024), an increasing number of studies are recognizing the importance of data diversity during training (Zhao et al., 2024; Miranda et al., 2023). However, diversity is a multifaceted concept, it encompasses variations in semantics, structures, and other properties. These different aspects can contribute differently to the model's learning process. Developing a principled understanding of these aspects and their roles is a key step toward building AI systems that are more data efficient.

Motivated by these challenges, we study data diversity through the lens of compositionality, which is described as "*infinite use of finite means*" (Chomsky, 2014), examining its impacts on model generalization. Specifically, we examine how different forms of diversity impact a model's ability to generalize compositionally; that is, to generalize to novel scenarios, by systematically recombining previously learned components and rules, which is referred to as compositional generalization. This capability is a hallmark of human cognition (Frankland & Greene, 2020; Dehaene et al., 2022; Elmoznino et al., 2024) and equally influential and important in AI, inspiring diverse research directions, including neuro-symbolic model (Soulos et al., 2023; 2024), disentangled representation learning (Wang et al., 2024b; Higgins et al., 2017), chain-of-thought reasoning (Wei et al., 2022), and some prompt engineering methods (Drozdov et al., 2022). A key advantage of compositional generalizing ability is that once a model learns to represent and manipulate the features in training data compositionally, it can generalize to novel combinations of those features without having

encountered them during training (Lake & Baroni, 2018; Mittal et al., 2021). This property is crucial for data-efficient learning (Wiedemer et al., 2023), enabling models to leverage a limited set of examples to generalize across a much larger combinatorial space.

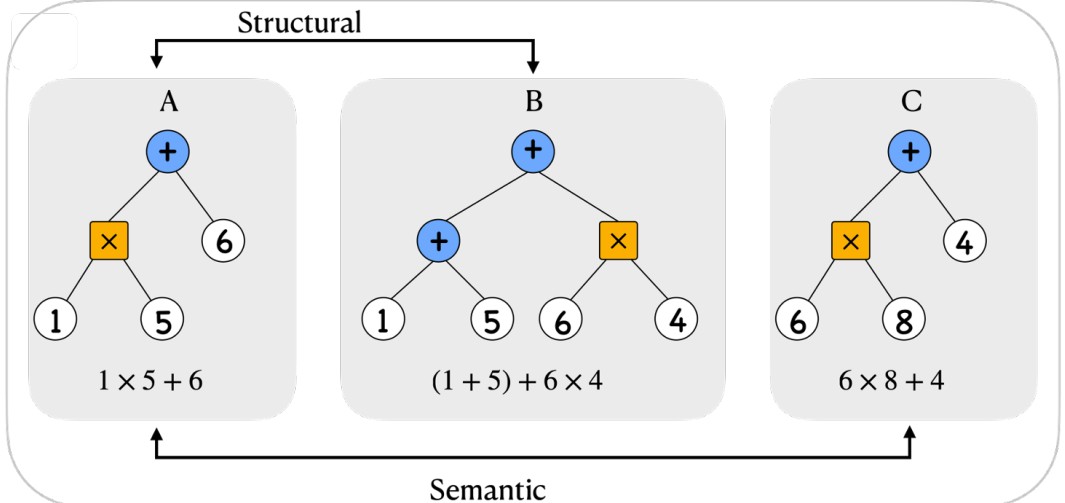

Figure 1: Example of semantic and structural diversity. In this algebraic circuit, A-B illustrates structural diversity: different circuit structures representing distinct compositional forms. In contrast, A-C illustrates semantic diversity: multiple surface realizations (different input values) that share the same underlying structure. The number of such realizations per structure reflects its degree of semantic diversity.

Training data plays a distinct role in enabling compositional generalization, since compositionality is a property of the data-generating process (Wiedemer et al., 2023; Zimmermann et al., 2021). Learning can be viewed as the inverse of this process, where the model infers underlying compositional components and then recombines these components to generalize beyond them. Prior work (Park et al., 2024; Okawa et al., 2023; Wiedemer et al., 2023) identifies two essential steps in achieving compositional generalization: (1) acquiring the fundamental conceptual components, and (2) systematically recombining them to generalize to novel compositions.

In this work, we study how data diversity, and its different aspects, support these processes under a constrained data budget. While prior work has explored the role of data diversity in compositional ability (Akyürek & Andreas, 2022; Zhou et al., 2023; Uselis et al., 2025), we formalize data diversity along two different dimensions: *semantic diversity*, which refers to varied surface realizations of the same underlying structure, and *structural diversity*, which captures a broader range of distinct compositional structures. Building on this distinction, we develop a theoretical framework to analyze how semantic and structural diversity contribute to models' compositional generalization ability under limited data. Our framework provides high-level guidelines for balancing these two dimensions of data diversity under a constrained data budget to maximize generalization. To validate our framework, we conduct controlled experiments on synthetic datasets of algebraic circuits (Wang et al., 2024a; Ito et al., 2024; Savage, 1998), which naturally supports decomposition and recombination of compositional structures. We then investigate how the theoretical insights and the controlled setting can be extended to realistic scenarios.

## 2 RELATED WORK

**Data Efficiency.** Large-scale datasets have enabled neural networks to achieve unprecedented success (Kaplan et al., 2020; Li et al., 2025). However, training on massive data comes at a high cost: it requires substantial computational resources, consumes substantial energy, and produces a massive amount of carbon footprint (Strubell et al., 2020; Patterson et al., 2021). Moreover, we are approaching practical and economic limits on collecting more data for training (Villalobos et al., 2022; Muennighoff et al., 2023). To address these challenges, a substantial body of work focused on data selection strategies (Xia et al., 2024; Xie et al., 2023; Gu et al., 2024; Albalak et al., 2024;

Yu et al., 2024). Most such approaches optimize data efficiency by selecting training examples according to various utility measures, e.g., quality and contribution to training loss reduction. Some studies suggest that data diversity can be equally important, but most existing work treats diversity as a broad and generic concept, typically defining and quantifying it in the embedding space using similarity-based measures (Friedman & Dieng, 2022; Miranda et al., 2023; Zhang et al., 2024). However, diversity can take multiple forms (Rahimi et al., 2023), such as semantic and structural diversity, which may influence not only how well a model fits the training set, but also how effectively it can extrapolate to novel combinations.

**Compositional Generalization.** Systematic compositionality is the ability of recombination of known parts and rules (Hupkes et al., 2020; Wiedemer et al., 2023), which is a feature of human cognition (Frankland & Greene, 2020; Dehaene et al., 2022; Elmoznino et al., 2024). Inspired by this, a growing body of work has emerged to analyze the compositionality of data (Wiedemer et al., 2023), the compositional generalization ability of machine learning models (Dziri et al., 2023; Ramesh et al., 2023; Lake & Baroni, 2023), as well as the process by which these abilities emerge during training (Park et al., 2024; Yang et al., 2024; Okawa et al., 2023). In particular, Park et al. (2024) and Okawa et al. (2023) used diffusion models to show that the emergence of compositional ability involved two stages. Model first disentangles individual components, and then it manipulates and recombines them to produce novel outputs. In this work, we focus on transformer-based models (Vaswani et al., 2017), as they form the backbone of most state-of-the-art AI systems. Although some studies have argued that transformers lack compositional generalization abilities (Dziri et al., 2023), these primarily focus on productive compositional generalization ability (Ramesh et al., 2023), which often requires generalizing to longer or more complex examples than those observed during training (Hupkes et al., 2020). By contrast, our work centers on systematic compositional generalization, emphasizing the model's ability to generalize to novel combinations of known components with known rules. Recent studies also investigated the impact of data diversity on compositional generalization, finding that increased diversity is beneficial (Zhou et al., 2023; Akyürek & Andreas, 2022; Uselis et al., 2025). Our work extends these findings by formally distinguishing and theoretically separating semantic diversity and structural diversity. This formal decomposition allows us to derive explicit generalization error bounds.

## 3 DATA-DRIVEN COMPOSITIONAL GENERALIZATION

To formalize data diversity from the perspective of compositionality, let the input space be $\mathcal{X}$ and output space be $\mathcal{Y}$, and $\mathbb{C}$ represents the set of compositional structures in training dataset. Each data instance can be disentangled into a structure and an input. A structure is denoted by $f_c$ (e.g., $x_1 \times x_2 + x_3$ in algebraic circuits as A and C in Figure 1), where $c \in \mathbb{C}$ specifies the composition. The inputs $\{x_{c,m}, \forall c \in \mathbb{C}, m \in [M]\}$ provide concrete instantiations of structures (e.g., $1 \times 5 + 6$ in Figure 1), where $[M] = \{1, 2, \cdots, M\}$, and yield data instances $(f_c(x_c, m), y_{c,m}) \in \mathcal{X} \times \mathcal{Y}$ (e.g., $(1 \times 5 + 6, 11)$). Furthermore, each structure $f_c$ can be decomposed into components. Formally, let $\mathcal{S}$ denote the pool of components, then the structure can be expressed as $f_c(x_{c,m}) = c(s_1(x_{c,m}), s_2(x_{c,m}), \cdots, s_t(x_{c,m}))$, where $s_i \in \mathcal{S}$ are the components, while the composition rule $c$ specifies how these components are combined. In algebraic circuits (Ito et al., 2024; Savage, 1998), for example, each $s_i(\cdot)$ may correspond to a sub-circuit (e.g., $x_1 \times x_2$).

The training dataset $\mathbb{D} = \{\{(f_c(x_{c,m}), y_{c,m})\}_{\forall m \in [M]}\}_{\forall c \in \mathbb{C}}$ consists of $N = |\mathbb{C}|$ distinct compositions, and each instantiated $M$ times with different inputs. The data budge is $K = M \cdot N$, corresponding to the number of training examples. In this framework, structural diversity is captured by $N$, while semantic diversity is captured by $M$. This setup explicitly connects data diversity to the two fundamental aspects of compositionality: components and combinations. Let $\mathbb{D}_c$ denote the set of all possible data consistent with the same underlying compositional structure $c$, and for the structure $c$, the training dataset has $\mathbb{D}'_c = \{(f_c(x_{c,m}), y_{c,m})\}_{\forall m \in [M]} \subset \mathbb{D}_c$, which is also a subset of the overall training dataset $\mathbb{D}$. The loss function $l(\cdot)$ is $L$-Lipschitz and bounded by $\lambda$.

### 3.1 ACQUIRE UNDERLYING COMPONENTS

An essential step toward compositional generalization is learning the components that constitute a structure, since these components form the basis of data-generating process Wiedemer et al. (2023). Given a dataset $\mathbb{D}'_c \subset \mathbb{D}_c$ consisting of $M$ different surface realizations of the same structure (semantic diversity), a model can decompose and learn the basic components (Park et al., 2024). The

generalization error on $\mathbb{D}_c$ primarily reflects the model's ability to learn these underlying components, since $f_c$ is fixed. One way to analyze the generalization error is through Rademacher complexity (Shalev-Shwartz & Ben-David, 2014), which provides a theoretical measure of the expressiveness of a hypothesis class and thus yields a bound on its generalization performance.

By standard results in statistical learning theory, the generalization error of a hypothesis class $\mathcal{H}$ within $\mathbb{D}_c$ is bounded in terms of its empirical error on $\mathbb{D}_c^{'}$, the empirical Rademacher complexity, and the number of training examples $M = |\mathbb{D}_c^{'}|$:

$$\text{err}_{\mathbb{D}_c}(h) \le \text{err}_{\mathbb{D}_c^{'}}(h) + 2 \cdot \text{Rad}_{\mathbb{D}_c^{'}}(l \circ \mathcal{H}) + 4\lambda\sqrt{\frac{2\ln 4/\delta}{M}}, \tag{1}$$

with probability of at least $1 - \delta$ for all hypotheses $h \in \mathcal{H}$. Here, $\text{err}_{\mathbb{D}_c}(h) = \mathbb{E}_{\mathbb{D}_c} l(h(f_c(x)), y_c)$ is the intra-compositional generalization error, $\text{err}_{\mathbb{D}_c^{'}}(h) = \frac{1}{M}\sum_{m=1}^{M} l(h(f_c(x_{c,m})), y_{c,m})$ is the empirical error on the training dataset, and $\text{Rad}_{\mathbb{D}_c^{'}}(l \circ \mathcal{H})$ is the empirical Rademacher complexity of the hypothesis class. This is detailed in Appendix B.

When the empirical error $\text{err}_{\mathbb{D}_c^{'}}(h)$ is small, e.g., when the model can fit the training dataset well, the intra-compositional generalization error can be bounded primarily by the Rademacher complexity and the number of training examples $M$. In particular, if $0 < \text{err}_{\mathbb{D}_c^{'}}(h) < \xi$ for any $\xi > 0$, we have:

$$\text{err}_{\mathbb{D}_c}(h) \le \xi + 2\text{Rad}_{\mathbb{D}_c^{'}}(l \circ \mathcal{H}) + 4\lambda\sqrt{\frac{2\log 4/\delta}{M}}. \tag{2}$$

Let $\mathcal{A} \subset \mathbb{R}^M$ denotes a set of vectors $\mathcal{A} = \{[h(f_c(x_{c,1})), \cdots, h(f_c(x_{c,M}))] \in \mathbb{R}^M | h \in \mathcal{H}\}$, which lies in an $r$-dimensional subspace of $\mathbb{R}^M$ and such that $\alpha = \max_{\boldsymbol{a} \in \mathbb{A}} ||\boldsymbol{a}||$. We use $r$ to characterize the semantic complexity, i.e., the internal variability of each component. For instance, in algebraic circuits, a deeper sub-circuit allows for a wider range of distinct surface realizations, leading to a larger $r$. Shown in Anthony & Bartlett (2009) and Shalev-Shwartz & Ben-David (2014), in this case, the Rademacher complexity satisfies:

$$\text{Rad}_{\mathbb{D}_c^{'}}(l \circ \mathcal{H}) \le \frac{6\alpha}{M}(\sqrt{r\log(2\sqrt{r})} + 2\sqrt{r}) = O(\frac{\alpha\sqrt{r\log r}}{M}). \tag{3}$$

Combining these results, we could yield an explicit upper bound of the intra-compositional generalization error in Theorem 3.1.

**Theorem 3.1** (Bound of Intra-Compositional Generalization Error). *Let $\mathcal{H}$ be a hypothesis class, and suppose the loss function $l(\cdot)$ is L-Lipschitz and bounded by $\lambda$. Assume the empirical loss satisfies $0 < err_{\mathbb{D}_c^{'}} < \xi$ for any $\xi > 0$. Let $\mathcal{A} \subset \mathbb{R}^M$ denotes a set of vectors, $\mathcal{A} = \{[h(f_c(x_{c,1})), \cdots, h(f_c(x_{c,M}))] \in \mathbb{R}^M | h \in \mathcal{H}\}$, lies in an $r$-dimensional subspace in $\mathbb{R}^M$, where $r$ characterizes the complexity of the underlying components, and $\alpha = \max_{\boldsymbol{a} \in \mathbb{A}} ||\boldsymbol{a}||$. Then, with probability at least $1 - \delta$, the intra-compositional generalization error for all hypotheses $h \in \mathcal{H}$ is bounded by:*

$$O(\frac{\alpha\sqrt{r\log r}}{M} + \lambda\sqrt{\frac{\log 1/\delta}{M}}).$$

## 3.2 GENERALIZE TO UNSEEN COMBINATIONS

Once a model has robustly learned individual components, a critical step is to learn to recombine them into novel compositional functions defined with respect to these components. This is often referred to as systematic compositional generalization (Hupkes et al., 2020). Achieving systematic compositional generalization requires learning to recombine components into novel functions. Since the process involves generalizing to unseen compositional structures, we bound this generalization error with covering number (Anthony & Bartlett, 2009) over the space of all possible compositions.

Assume that all possible compositional functions in $\mathbb{C}_{\text{all}}$ (the set of compositional functions in the training dataset is a subset $\mathbb{C} \subset \mathbb{C}_{\text{all}}$) can be embedded into a $d$-dimensional space $\mathcal{V}$ by an embedding function $\phi(\cdot)$, where each point in $\mathcal{V}$ (i.e., $\phi(c^{'})$ and $c^{'} \in \mathbb{C}_{\text{all}}$) represents a specific combination of learned components, and distances between points in the space reflect the similarity between compositions. The space $\mathcal{V}$ is modeled as a $d$-dimensional Euclidean space, which is applied to the latent

representation $\phi(\cdot)$ and enables tractable geometric analysis. We use $d$ to characterize the structural complexity, i.e., the diversity of possible combinations of components. For example, a composition with more layers allows for a greater combinatorial richness of arrangements, resulting in a larger $d$. Under this assumption, if a model $\mathcal{M}_\theta$ (e.g., a hypothesis $h \in \mathcal{H}$) is trained on a dataset containing a particular compositional function $c \in \mathbb{C}$, it is expected to generalize more easily to similar compositions, i.e., those $c^{'} \in \mathbb{C}_{\text{all}}$ satisfying $||\phi(c) - \phi(c^{'})|| \leq \epsilon$ for some threshold $\epsilon > 0$. Furthermore, the generalization performance on $c^{'}$ degrades with $||\phi(c) - \phi(c^{'})||$. For example, suppose the training composition corresponds to the circuit $c_1(\cdot) = ((x_1 + x_2) + x_3) \cdot x_4$, and a structurally similar variant such as $c_2(\cdot) = (x_1 + (x_2 + x_3)) \cdot x_4$ satisfies $|\phi(c_1) - \phi(c_2)| \leq \epsilon$. In contrast, a structurally distant variant such as $c_3(\cdot) = (x_1 + x_2) \cdot (x_3 + x_4)$ satisfies $|\phi(c_1) - \phi(c_3)| > \epsilon$. Although the true conceptual space may be more complex, modeling it as a $d$-dimensional Euclidean space enables tractable geometric analysis. In this view, each composition observed during training defines an $\epsilon$-radius ball in the space $\mathcal{V}$, representing the region over which $\mathcal{M}_\theta$ can generalize from $c \in \mathbb{C}$ with generalization error smaller than $\epsilon$, i.e., $\text{err}_c(c^{'}) \leq \epsilon$, where $\text{err}_c(c^{'}) = \mathbb{E}_{\mathbb{D}_{c'}} l(\mathcal{M}_\theta(f_{c'}(\cdot)), y_{c'})$. So, the radius $\epsilon$ serves as an upper bound of inter-compositional generalization error.

Training on $N = |\mathbb{C}|$ distinct compositions corresponds to covering the space with $N$ $\epsilon$-radius balls. As $N$ increases, the space $\mathcal{V}$ can be covered using balls with smaller radius $\epsilon$, implying that a larger training set permits finer granularity in coverage and reduces the upper bound of inter-compositional generalization error. Conversely, when $N$ is small, a larger $\epsilon$ is required for coverage, which leads to higher inter-compositional generalization error. Minimizing $N$ while keeping $\epsilon$ small captures the trade-off between data efficiency and inter-compositional generalization in our framework. Formally, the covering number $(N(\epsilon, \mathcal{V}, || \cdot ||))$ is defined as the minimal number of balls (under norm $|| \cdot ||$) required to cover the space $\mathcal{V}$. This notion can be used to establish a relationship between the error bound $\epsilon$ and the number of distinct compositions $N$. This is detailed in Appendix C.

From standard results in statistics (Shalev-Shwartz & Ben-David, 2014; Mohri et al., 2018), the covering number $N$ can be bounded as a function of the space dimension $d$, which could be used to characterize the compositional complexity, and the radius $\epsilon$ as $N \leq (\frac{2}{\epsilon} + 1)^d$. This could be rearranged to express the effect of number of compositions provided in training set, i.e., $N$, on the upper bound of the inter-compositional generalization error, i.e., $\epsilon$:

$$\text{err}_{\mathbb{C}_{\text{all}}} \leq \epsilon \leq \frac{2}{N^{\frac{1}{d}} - 1}. \tag{4}$$

We formalize this in Theorem 3.2.

**Theorem 3.2** (Bound of Inter-Compositional Generalization Error). *Let the set of all possible compositions $\mathbb{C}_{\text{all}}$ form a conceptual space $\mathcal{V}$, modeled as a $d$-dimensional Euclidean norm ball, where $d$ characterizes the compositional complexity. Assume each composition observed during training covers an $\epsilon$-radius ball in this space, within which the inter-compositional generalization error $\text{err}_{\mathbb{C}_{\text{all}}}$ is at most $\epsilon$. Then, the inter-compositional generalization error can be bounded by:*

$$O(\frac{1}{N^{\frac{1}{d}} - 1}),$$

*where $N$ is the number of distinct compositions presented in the training data.*

### 3.3 Role of Data diversity in compositional generalization

We established upper bounds for both intra-compositional and inter-compositional generalization error in Theorem 3.1 and Theorem 3.2, respectively, and the results connect directly to the two dimensions of data diversity discussed earlier. The overall error bound of generalization error from the perspective of compositions can be expressed as the summation of these two:

$$\text{err} = O(\frac{\alpha\sqrt{r\log r}}{M} + \lambda\sqrt{\frac{\log 1/\delta}{M}} + \frac{1}{N^{\frac{1}{d}} - 1}). \tag{5}$$

Here, the first two terms depend on semantic diversity through $M$, capturing the model's ability to generalize across variations of the same underlying structure (e.g., A-C in Figure 1); while the last term depends on structural diversity through $N$, capturing its ability to generalize to new combinations (e.g., A-B in Figure 1). We combine these two error bounds by addition, reflecting the

decomposition where minimizing the total error requires robust component learning and effective generalization across structures. Achieving zero error in one component does not imply perfect generalization overall, justifying the additive model. For simplicity, we omit potential coupling terms in our analysis, and detailed discussion is provided in Appendix A.

Given a fixed data budget $K = M \cdot N$, the overall generalization error could be upper bounded by the term of $N$, the number of distinct compositions exposing during training:

$$\text{err} = O(\mathcal{F}(N, r, d)), \quad \text{s.t. } \mathcal{F}(N, r, d) = O\left(\frac{N\alpha\sqrt{r\log r}}{K} + \lambda\sqrt{\frac{N\log 1/\delta}{K}} + \frac{1}{N^{\frac{1}{d}} - 1}\right). \quad (6)$$

Equation 6 illustrates that different aspects of data diversity affect the generalization error bound in opposite ways under a fixed data budget. Specifically, in the first two terms, increasing $N$ enlarges the error bound. In contrast, the last term decreases with $N$, as additional compositions improve structural coverage. This demonstrates that increasing diversity without a principled strategy does not necessarily lead to optimal generalization ability, and the training dataset requires a balanced allocation between semantic and structural diversity to achieve data efficiency.

To further investigate how different aspects of diversity influence the generalization performance, we differentiate the bound in Equation 6 with respect to $N$. This yields:

$$\frac{\partial \mathcal{F}(\cdot)}{\partial N} = \frac{\alpha\sqrt{r\log r}}{K} + \frac{\lambda\zeta}{2\sqrt{\zeta N}} - \frac{N^{\frac{1}{d}-1}}{d(N^{\frac{1}{d}} - 1)^2}, \quad \text{s.t. } \zeta = \frac{\log 1/\delta}{K}. \quad (7)$$

As illustrated in Figure 2, once $N$ exceeds a point $N^*$, introducing additional distinct compositions leads to an increase in the error bound. Moreover, both the component complexity (captured by $r$) and the compositional complexity (captured by $d$) have a significant effect on the derivative and thus influence the increasing or decreasing trend of the error bound. While the stationary points $N^*$ may appear close in the illustrative figure, in practice different $r$ and $d$ values can induce substantial shifts in $N^*$. These results are summarized formally in Theorem 3.3.

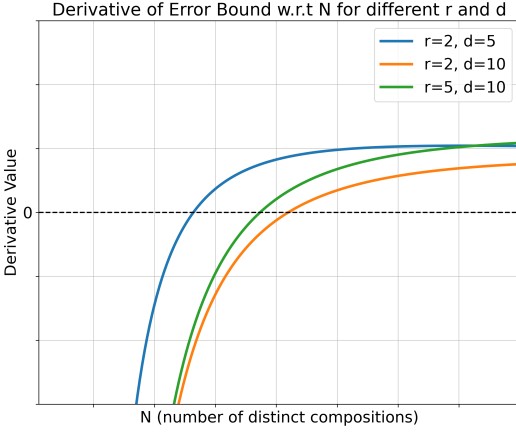

Figure 2: Derivative of the generalization error bound with respect to $N$ under varying levels of component complexity $r$ and compositional complexity $d$. When the derivative is negative, adding more distinct compositions $N$ decreases the error bound, whereas once it becomes positive, further increasing $N$ causes the error bound to rise. Both $r$ and $d$ influence the threshold (stationary point) at which the transition occurs.

**Theorem 3.3** (Monotonicity of the Generalization Error Bound). *The bound $\mathcal{F}(N, r, d)$ be defined as in Equation 6 under a fixed data budget $K = M \cdot N$. Here, $r$ characterizes the component complexity and $d$ characterizes the compositional complexity. Then, the following monotonicity properties hold with respect to $N$: (i) There exists a stationary point $N^* > 1$, such that $\mathcal{F}(N, r, d)$ is strictly decreasing for $N < N^*$ and strictly increasing for $N > N^*$. (ii) The stationary point $N^*$ decreases as $r$ increases, i.e., higher component complexity. More semantic diversity will provide more benefits when the dataset has higher component complexity. (iii) The stationary point $N^*$ increases as $d$ increases, i.e., greater compositional complexity and lower component complexity. More structural diversity will provide greater benefits when the dataset has higher compositional complexity.*

Our analysis does not aim to determine the exact value of the optimal $N^*$, but rather to characterize the monotonic behavior and provide high-level guidance for allocating the two dimensions of data diversity under constrained data budget. Specifically, when $N^*$ is small, which corresponds to high component complexity, the regime $N > N^*$ is more prevalent. In this case, increasing semantic diversity, i.e., providing more realizations per composition structure, contributes more effectively. In contrast, when $N^*$ is large, corresponding to low component complexity and a compositionality bottleneck, the regime

$N < N^*$ is more common. In this case, increasing structural diversity, i.e., exposing more distinct compositions, has a greater impact on improving generalization.

# 4 EMPIRICAL EVALUATION

We empirically evaluate our theoretical framework, beginning with fully controlled synthetic datasets to real-world data. Our goal is to investigate whether the two dimensions of data diversity introduced in our framework manifest in practice, and in particular, to validate whether the proposed allocation strategy leads to improved generalization or more data-efficient training. We emphasize that this approach is fundamentally different from existing data selection or argumentation strategies. Our framework, in contrast, proposes structural design principles based on inherent task properties and diversity. We begin with synthetic algebraic circuits, employing both a small-scale encoder-only model and large language models (LLMs) such as GPT-2-XL (Brown et al., 2020) and Mistral 7B v0.1 (Jiang et al., 2023). This experimental design enables precise control over data diversity, allowing us to examine how models behave under different allocations of semantic and structural diversity, and it further highlights the role of model capacity and task difficulty affect the results, corresponding to the component complexity and compositional complexity introduced in Theorem 3.3. We then validate our framework on a real-world dataset GSM8K (Cobbe et al., 2021), which can be naturally represented as circuit structures, using Mistral 7B v0.1 (Jiang et al., 2023) to assess our findings under realistic settings. Although the models we employ are not among the most recent, we deliberately choose them because state-of-the-art models already achieve strong performance, making additional fine-tuning or training less informative for assessing the impact of data diversity.

## 4.1 SYNTHETIC SETUP

This setting provides full controllability over both semantic and structural diversity, enabling a direct test of our theoretical framework. Each circuit corresponds to an algebraic expression such as $4 \times (5 \times 2 + 5)$. By computing the result modulo 10, the problem becomes a modular arithmetic. We further introduce variability through sub-circuits to control the component complexity. By adjusting circuit structure, sub-circuit depth, number of distinct structures ($N$), and number of realizations of the same underlying structure ($M$), we can systematically vary both semantic and structural diversity, as well as the component and structural complexity. The dataset is constructed by $K$ training examples, where $K = M \cdot N$. Thus, structural diversity is determined by $N$, while semantic diversity is determined by $M$. This is detailed in Appendix G. We consider two types of model here: (i) training an encoder-only model from scratch, and solve each circuit as a 10-classes classification problem (ii) fine-tune a pretrained large language model, where both the final answer and the chain-of-thought (CoT) are provided in the training dataset, an example is shown in Appendix D. This effectively guides the model to perform an explicit decomposition during training.

### 4.1.1 ENCODER-ONLY MODEL

We first use an encoder-only model trained from scratch on the synthetic dataset. The model is implemented as a one-layer transformer block with a standard embedding layer, a multi-head self-attention layer, and a feed-forward layer. The model's architecture and the training hyperparameters are detailed in Appendix D.1. We monitor the loss on the testing set, in which each circuit is composed of the same set of sub-circuits as those in the training dataset, but arranged into novel compositional structures. This design enables us to evaluate the model's ability to generalize to novel structures composed of previously learned components.

Figure 3 shows evaluation loss curves of the one-layer transformer under different allocations of semantic and structural diversity ($M$ and $N$), with a fixed data budget $K = 10,000$. When the components are simple ($r$ is small; e.g., 1-layer sub-circuit and overall depth 3), generalization is primarily constrained by recombination. In this regime, the stationary point $N^*$ in the error bound becomes large, meaning that models often operate in the range $N < N^*$. As a result, increasing structural diversity (i.e., raising N) leads to improved performance, while models trained with limited structural diversity tend to plateau at higher loss. In contrast, when components are more complex ($r$ is large; e.g., 2-layer sub-circuit with overall depth 3), the regime $N > N^*$ becomes more likely, and the benefits of increasing structural diversity diminish or even disappear, as reflected by the narrowing of loss curves across different values of $N$. This indicates that as sub-

circuits become harder to learn, the bottleneck shifts from recombination to component acquisition, and the stationary point $N^*$ shifts left as in Figure 2. Taken together, these results highlight a key aspect of our theoretical framework: a model's ability to achieve systematic compositional generalization depends on both acquiring individual components and learning how to recombine them.

When components are relatively simple, greater structural diversity (i.e., more distinct compositions) is beneficial, as the challenge lies in learning how to recombine components. Conversely, when components are more complex, increasing semantic diversity (i.e., providing more surface realizations per composition) becomes more effective, as the bottleneck lies in learning the components themselves.

### 4.1.2 LARGE LANGUAGE MODELS

We evaluate LLMs, specifically GPT-2-XL (1.5B) (Brown et al., 2020) and Mistral 7B v0.1 (7B) (Jiang et al., 2023), on the synthetic algebraic circuits dataset. For these models, we additionally provide CoT reasoning traces for solving the problems, as shown in Appendix D. Using LLMs serves two purposes: (i) to examine whether our theoretical framework holds when applied to large-scale models, and (ii) to investigate how model capacity influences these insights. In particular, we study how the component complexity ($r$ in Theorem 3.3) shifts as

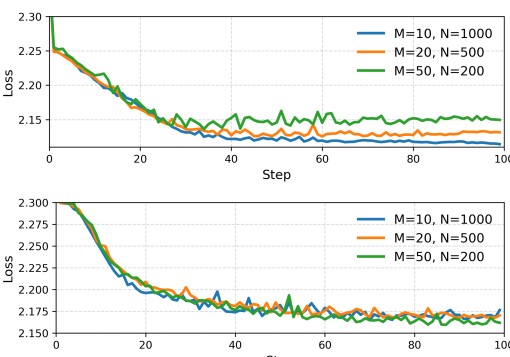

Figure 3: *Top*: Evaluation loss curve for circuits with 1-layer sub-circuits and overall depth 3. Structural diversity plays a dominant role, with higher $N$ (number of distinct compositions) yielding lower loss. *Bottom*: Evaluation loss curve for circuits with 2-layer sub-circuits and overall depth 3. As component complexity increases, the performance gap across different values of $N$ narrows.

model capacity increases. Intuitively, higher-capacity models are able to solve the same components with greater ease, such that component complexity decreases. This allows us to connect the theoretical framework to pretrained models, thereby providing evidence that our findings extend beyond small-scale settings. These models may already encode compositional patterns and reasoning skills due to their large-scale pretraining. This prior knowledge could potentially reduce the marginal benefit of additional diversity in the fine-tuning stage. However, we deliberately chose to use non-state-of-the-art models for fine-tuning because they offer a clearer window into observing the impact of the data diversity on generalization.

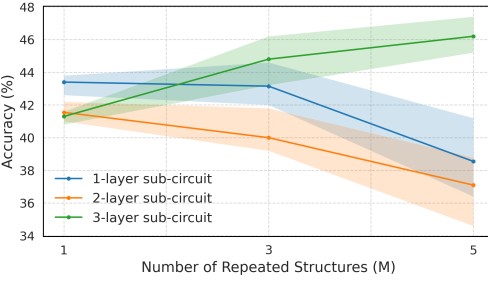

Figure 4: Performance of GPT-2-XL fine-tuned on algebraic circuits with different levels of semantic diversity ($M$). Results are shown for circuits of overall depth 4. Increasing semantic diversity improves accuracy when component complexity is high, but can degrade performance at low sub-circuit complexity.

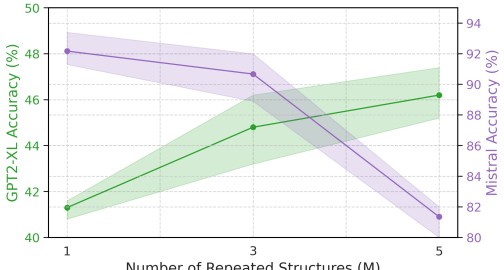

Figure 5: Performance of GPT-2-XL (1.5B) and Mistral (7B) fine-tuned on algebraic circuits with varying levels of semantic diversity ($M$). For GPT-2 XL, increasing semantic diversity improves performance, whereas the opposite trend is observed for Mistral: higher semantic diversity degrades performance.

In Figure 4 and Figure 5, We report 5 independent runs, including data sampling, training and inference. Solid lines denote the average performance, and shaded regions indicate the range across runs. Figure 4 shows the test accuracy of GPT-2-XL fine-tuned with a fix data budget $K = 4,500$ with CoT, tested on a set of 500 distinct circuits that do not overlap with the training data. When

the components are relatively simple ($r$ is small, e.g.,1-layer and 2-layer sub-circuits), increasing semantic diversity (i.e., increasing $M$ and decreasing $N$) offers no benefits and may even reduce performance, as the stationary point $N^*$ is large and $N$ lies below the threshold, i.e., $N < N^*$ with high probability. In contrast, when the components are more difficult ($r$ is large, e.g., 3-layer sub-circuits), increasing semantic diversity yields better performance, as the stationary point $N^*$ shifts to a smaller value, and models typically operate in the regime $N > N^*$, where additional semantic diversity is more beneficial. Thus decreasing $N$ moves the allocation towards $N^*$ and improves performance. Next, we extend our study on Mistral (7B), which is substantially larger, and more powerful than GPT-2-XL. We fine-tune Mistral using the dataset with 3-layer sub-circuits. As shown in Figure 5, increasing semantic diversity harms performance for Mistral. Compared to GPT-2-XL, Mistral's greater capacity suggests that each component is easier for it to learn, i.e., the component difficulty $r$ is lower for Mistral. According to Theorem 3.3, as $r$ decreases, the stationary point $N^*$ increases and shifts to larger values. Under a fixed training budget $K$, increasing $M$ and reducing $N$, shifts the data allocation away from the optimal, leading to degraded performance. These results indicate that simply increasing one form of diversity does not guarantee optimal performance; instead, the ideal allocation depends on both the task characteristics and the model's capacity. This further demonstrates that our theoretical framework offers practical guidance for fine-tuning large, pretrained models.

## 4.2 Real Data

To validate our framework beyond synthetic settings, we apply it to real-world data. Mathematical problems can naturally be abstracted into circuit-like structures as detailed in Appendix F, where each problem is decomposed into functional components. The problems in GSM8K dataset are essentially shallow circuits: most questions require only one or two layers of composition, but solving each sub-step is non-trivial for models, as it involved not only carrying out basic operations but also understanding and formalizing the problem statement. In our framework, this corresponds to low compositional complexity ($d$ is small) and high component complexity ($r$ is large), suggesting that semantic diversity is more critical, since deep compositional reasoning is rarely required.

We use GPT-4o (Hurst et al., 2024) to generate GSM-style examples. The generated examples follow the same underlying reasoning paths as the original one in GSM8K dataset but differ in their surface realization, thereby enriching se-

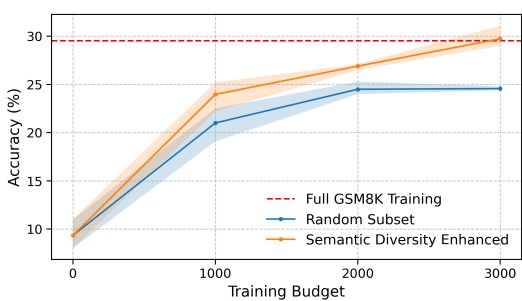

Figure 6: Model performance under different training budgets. We compare randomly sampled subsets of GSM8K to a semantic diversity-enhanced setting, where each seed example is paired with a semantically diverse variant of identical reasoning structure. We report the results of 4 independent runs, including data sampling, training, and inference. Solid lines denote the average performance, and shaded regions indicate the range across runs.

mantic diversity without altering compositional structure. The prompts used for generating GSM-style examples, and example outputs are provided in Appendix E. We then fine-tune Mistral on two different data configurations: (i) a random subset, in which we use examples randomly sampled from GSM8K (resulting in higher structural diversity, i.e., larger $N$), and (ii) a semantic diversity enhanced dataset, in which we have pairs data where each GSM8K seed example is accompanied by a semantically diverse variant with identical reasoning structure (smaller $N$), as shown in Appendix E. The structural diversity is intentionally reduced in configuration (ii), allowing us to study the impact of this shift in diversity allocation. In both cases, we control the total training budget. For example, if the budget $K = 5,000$, configuration (i) uses $5,000$ randomly selected GSM8K problems, while configuration (ii) uses $2,500$ seed examples and $2,500$ GPT-4o-generated variants.

Figure 6 compares the performance of fine-tuned models under different training data budget. Across all budget levels, subsets curated to enhance semantic diversity consistently outperform size-matched random subsets. Remarkably, with only $3,000$ training examples, the semantic diversity-enhanced setting matches or even exceeds the accuracy achieved by fine-tuning on the full GSM8K dataset. In this regime, components complexity is relatively high ($r$ is large), while compositional

complexity remains low ($d$ is small). According to Theorem 3.3, the stationary point $N^*$ is therefore small. Reducing structural diversity ($N$) in this context moves the diversity allocation closer to $N^*$, leading to improved performance. This finding suggests that identifying an effective balance between semantic and structural diversity can significantly enhance a model's ability to generalize to unseen data in a data-efficient manner.

## 5 CONCLUSION

In this work, we analyzed the role of data diversity through the lens of compositional generalization and explore how it can enable more data-efficient learning. We refined the broad concept of diversity into two complementary dimensions: semantic diversity, which supports the acquisition of underlying components, and structural diversity, which enables generalization to unseen compositions. Our theoretical framework demonstrates that these two dimensions influence generalization in different ways, and further reveals that under a constrained data budget, the optimal balance between them depends on both component complexity and compositional complexity. Experiments on synthetic datasets validated the framework and highlighted the role of model capacity in shifting the optimal allocation. Results on GSM8K further confirmed the framework's applicability to real-world problems. Overall, our findings highlight the critical role of data diversity and provide a principled strategy for constructing datasets that enable more data-efficient and generalizable training. A direction for future research is to extend this theoretical framework to other mathematical and logical settings, incorporating the new diversity dimensions and types of compositionality that those domains would naturally introduce.

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

ACKNOWLEDGEMENT

The authors used large language models (LLMs) as tools to assist with polishing the writing of this manuscript. The use of LLMs was limited to improving clarity and readability; all technical content, ideas, analyses, and conclusions are work of the authors.

## A   LIMITATION

While our work offers a theoretical and empirical analysis of how data diversity affects model generalization, it has several limitations. First, our framework treats semantic diversity and structural diversity as independent factors, ignoring their potential interactions. In practice, such interactions likely exist and may influence the optimal balance between the two. Second, we do not explore alternative training strategies, such as explicitly decomposing problems into subproblems and training models on them or curriculum training (Soviany et al., 2022). Our analysis is restricted to settings where such decomposition is not feasible. Finally, we do not provide concrete methods to measure semantic diversity, structural diversity, component complexity, or compositional complexity. As a result, our contributions remain a high-level principles rather than operational metrics.

## B   RADEMACHER COMPLEXITY

Uniform convergence is a sufficient condition for learnability, and Rademacher complexity measures the rate of uniform convergence. We define a hypothesis class $\mathcal{H}$, loss function $l$, the set of all possible input-output pairs consistent with the same underlying compositional structure $\mathbb{D}_c$ and the training dataset $\mathbb{D}'_c = \{(f_c(x_{c,m}), y_{c,m}) | m \in [M]\} \subset \mathbb{D}_c$. To simplify notation, define the function class:

$$\mathcal{G} := l \circ \mathcal{H}.$$

For any $g \in \mathcal{G}$ and corresponding $h \in \mathcal{H}$, define the generalization and empirical errors as:

$$\text{err}_{\mathbb{D}_c}(h) = \mathbb{E}_{\mathbb{D}_c}[g(f_c(\cdot), y_c)], \quad \text{err}_{\mathbb{D}'_c}(h) = \frac{1}{M} \sum_{m=1}^{M} g(f_c(x_{c,m}), y_{c,m}).$$

Let $\sigma$ be a vector of independent and identically distribution (i.i.d.) Rademacher variable such that $P[\sigma_m = 1] = P[\sigma_m = -1] = 0.5$. The empirical Rademacher complexity of $\mathcal{G}$ with respect to dataset $\mathbb{D}'_c$ is defined as follows:

$$\text{Rad}_{\mathbb{D}'_c}(\mathcal{G}) = \frac{1}{M} \mathbb{E}_{\sigma \sim \{\pm 1\}^M}[\sup_{g \in \mathcal{G}} \sum_{m=1}^{M} \sigma_m g(f_c(x_{c,m}), y_{c,m})].$$

More generally, given a set of vectors, $\mathcal{A} \subset \mathbb{R}^M$, the Rademacher complexity of $\mathcal{A}$ is:

$$\text{Rad}(\mathcal{A}) = \frac{1}{M} \mathbb{E}_{\sigma \sim \{\pm 1\}^M}[\sup_{a \in \mathcal{A}} \sum_{m=1}^{M} \sigma_m a_m].$$

Assuming the loss function $l(\cdot)$ is bounded by a constant $\lambda$, for all $h \in \mathcal{H}$ we have:

$$\text{err}_{\mathbb{D}_c}(h) - \text{err}_{\mathbb{D}'_c}(h) \leq 2\text{Rad}_{\mathbb{D}'_c}(l \circ \mathcal{H}) + 4\lambda \sqrt{\frac{2 \ln(4/\delta)}{M}},$$

with probability of at least $1 - \delta$. This is result in derived using standard uniform coverage arguments (Shalev-Shwartz & Ben-David, 2014).

## C   TRADE-OFF BETWEEN DATA EFFICIENCY AND GENERALIZATION ERROR

Assume that all possible combinations can be embedded in a $d$-dimensional space $\mathcal{V}$ by an embedding function $\phi(\cdot)$, where each composition corresponds to a point in this space. The model $\mathcal{M}$ is

Figure 7: Covering a compositional space with $\epsilon$-radius balls. Each point represents a composition observed in the training set, and the model is expected to generalize to nearby points. Increasing the number of training examples reduces the radius $\epsilon$, yielding finer coverage of the space. Conversely, with fewer training examples, each ball must have larger radius to cover the space, leading to higher generalization error.

expected to generalize to similar compositions, i.e., nearby points in the $d$-dimensional space. As shown in Figure 7, the model can generalize to points with distance at most $\epsilon$ centered at each training composition with acceptable error. For example, if the training composition corresponds to the circuit $c_1(\cdot) = ((x + y) + z) \cdot w$, then a structurally nearby variant such as $c_2(\cdot) = (x + (y + z)) \cdot w$ falls in the $\epsilon$-radius ball (i.e., $|\phi(c_1) - \phi(c_2)| \leq \epsilon$). In contrast, a structurally far variant such as $c_3(\cdot) = (x + y) \cdot (z + w)$ would lie outside the $\epsilon$-radius ball (i.e., $|\phi(c_1) - \phi(c_3)| > \epsilon$).

Generalization, in this view, corresponds to covering the entire conceptual space from the sampled compositions, where the radius $\epsilon$ reflects the upper bound of generalization error. When the training dataset contains only a few compositions, covering the whole space requires each ball to have a larger radius (i.e., larger $\epsilon$), which implies higher generalization error. In contrast, when the training set includes more diverse compositions, the space can be covered more densely using many smaller balls, each with a smaller radius (i.e., smaller $\epsilon$), leading to lower generalization error.

## D    CHAIN-OF-THOUGHT OF SOLVING ALGEBRAIC CIRCUIT

Below is an example of the CoT format used to fine-tune LLMs on the algebraic circuit tasks.

---

**Example 1: CoT Example**

**Question:** $4 \times (5 \times 2 + 5)$

**CoT:** We solve this step by step:

1. $4 \times (10 + 5)$

2. $4 \times 15$

3. $60$

4. $60$ Mod $10$, the final answer is $0$.

---

Table 1: Hyperparameters of the one-layer transformer model used in our experiments.

| Layer | Output Size | Details |
|---|---|---|
| Input Embedding | 32 | Token embedding + positional encoding |
| Transformer Block | 32 | One-layer transformer block |
| Attention heads | 32 | 16 attention heads |

Table 2: Training hyperparameters for the encoder-only classification model.

| Parameter | Value |
|---|---|
| Batch size | 128 |
| Learning rate | $1 \times 10^{-3}$ |
| Weight decay | 0.01 |
| Max sequence length | 256 |
| Number of epochs | 100 |

## D.1 ENCODER-ONLY MODEL

We use a single layer encoder-only transformer as the classification model. The architecture and training hyperparameters are summarized in Table 1 and Table 2, respectively.

## E GENERATE GSM-STYLE EXAMPLES

We use GPT-4o to generate GSM-style questions and answers, including the question, CoT reasoning, and final answer. To ensure the generated examples follow the same reasoning path as seed examples from GSM8K, we use a prompt that preserves the logical structure while altering the surface realization (e.g., names, numbers, and context). The prompt used is shown bwlow:

---

**Example 2: Prompt**

You are given a GSM problem and its full solution. Your task is to generate a similar question:
1. Has a different surface realization (different names, quantities, etc.).
2. Has a correct answer.
3. Requires the same reasoning process as the original one.
4. Produces a full GSM-style answer:
- Includes reasoning steps
- Give the final answer using the format: $\#\#\#$ answer

Original Problem: {original question}
Original Answer: {original answer}
Now write the question and answer pair please:
Output:
Question: ...
Answer: ...

---

An example pair of GSM8K seed example and the generated example is provided:

---

**Example 3: Pair of example**

**Seed Example:**

*Question:* Natalia sold clips to 48 of her friends in April, and then she sold half as many clips in May. How many clips did Natalia sell altogether in April and May?

*Answer:* Natalia sold $48/2 =<< 48/2 = 24 >>$ 24 clips in May. Natalia sold $48 + 24 =<< 48 + 24 = 72 >>$ 72 clips altogether in April and May. $\#\#\# 72$

**Generated Example:**

*Question:* Daniel baked 48 cookies in the morning, and then he baked twice as many cookies in the afternoon. How many cookies did Daniel bake altogether in the morning and afternoon?

*Answer:* Daniel baked $48 \times 2 =<< 48 * 2 = 96 >>$ 96 cookies in the afternoon. Daniel baked $48 + 96 =<< 48 + 96 = 144 >>$ 144 cookies altogether in the morning and afternoon. $\#\#\# 144$

---

## F  GSM PROBLEM AS CIRCUIT

Mathematical problems can be naturally abstracted into circuit structures and by decomposing them into functional components, as shown in Figure 8. For example, the problem in GSM8K "*Natalia sold clips to 48 of her friends in April, and then she sold half as many clips in May. How many clips did Natalia sell altogether in April and May?*" can be decomposed into computing the number of clips sold in May and summing the totals from April and May. Each component corresponds to a node in the circuit, and the entire problem forms a compositional structure analogous to an algebraic circuit.

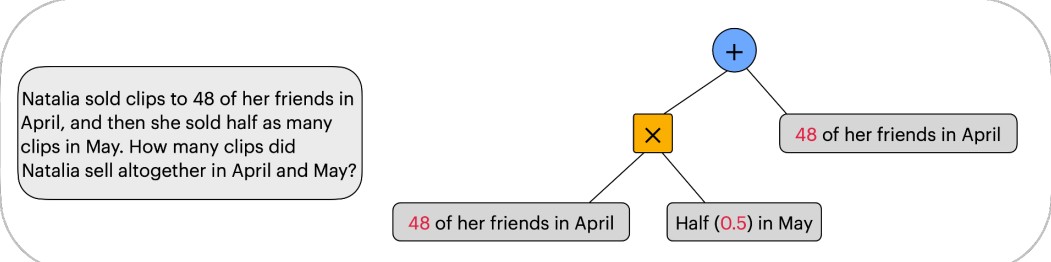

Figure 8: Decomposition of a GSM8K problem with circuit. The problem is represented as a composition of functional nodes: one computes May sales $48 \times 0.5 = 24$, and another combines April and May sales via $48 + 24 = 72$ to yield the final answer.

## G  SYNTHETIC SETUP

In the synthetic algebraic circuit dataset, each circuit corresponds to an algebraic expression such as $4 \times (5 \times 2 + 5)$. By computing the result modulo 10, the problem becomes a modular arithmetic, where the model needs to predict the output value between 0 and 9. We enforce the dataset to be balanced across classes, since otherwise results equal to 0 would dominate disproportionately. We further introduce variability through sub-circuits, where a full circuit can be composed of smaller sub-circuit as shown in Figure 9, which illustrated how sub-circuits are reused across compositions. We ensure that circuits in both the training and testing sets to be constructed from the same set of components. Increasing the depth of sub-circuits raises the component complexity. By adjusting circuit structure, sub-circuit depth, number of distinct structures ($N$), and number of realizations

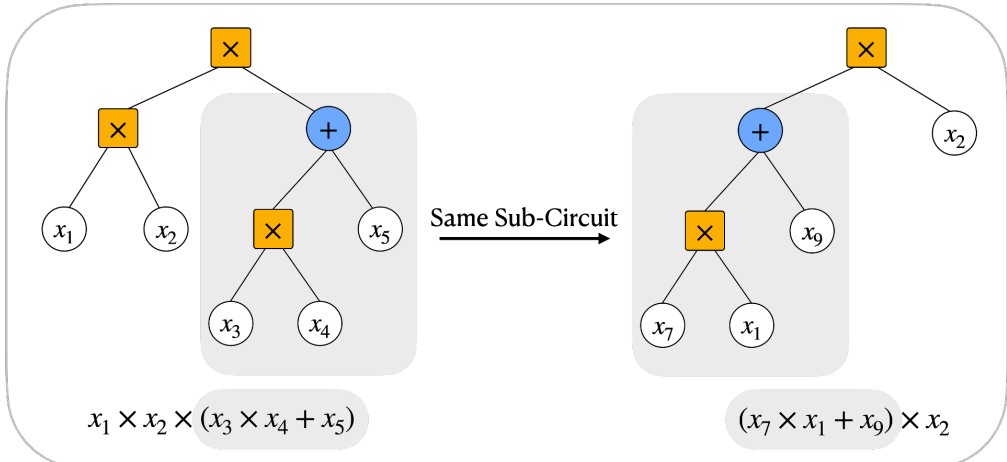

Figure 9: Example of basic component sharing in algebraic circuits. Each full circuit is composed of sub-circuits (components), which may be reused across different compositions. The depth of a sub-circuit determines the component complexity, while the way components are combined controls the overall structural diversity.

of the same underlying structure ($M$), we can systematically vary both semantic and structural diversity, as well as the component and structural complexity. We construct the dataset under a fixed budget of $K$ training examples. Specifically, we select $N$ distinct circuit structures, and generate $M$ surface realizations for each structure, ensuring $K = M \cdot N$. In this way, structural diversity is determined by $N$, while semantic diversity is determined by $M$.

