# OpenReview forum: "Data Diversity for Compositional Generalization"
_ICLR.cc/2026/Conference — Submitted to ICLR 2026_

### Official Review · Reviewer_JbLv · 2025-10-22

**Soundness:** 3
**Presentation:** 3
**Contribution:** 3
**Rating:** 4
**Confidence:** 4

**Summary:**

This paper studies the two data dimensionalities of semantic diversity and structural diversity individually, with Rademacher complexity and covering numbers as theoretical tools, respectively. The experiments on synthetic datasets and real-world datasets with fixed data budget confirm the theoretical findings and provide insights like the importance of a principled strategy with respect to data diversity to improve compositional generalization.

**Strengths:**

- The theoretical findings, e.g., bounds of generalization, are sound. The formulations are well-defined and tools are established (i.e., Rademacher complexity and covering numbers), though there are some typos.
- The experiments on datasets, including synthetic datasets and real-world datasets, are comprehensive and the conclusions are insightful.

**Weaknesses:**

- The theoretical results still hinge on the validity of the assumptions. The embedding of all compositional functions into a d-dimensional Euclidean space. How well does this idealized geometry capture the true "conceptual space" of real-world problems? If the training structural combinations are confined to a narrow space in this d-dimensional space, the generalization performance is expected to be very low (i.e., a large radius $\epsilon$ to cover all possible combinations).
- The scope of this work is limited to algebraic circuits. Thus, only structural and semantic diversities are considered, which, based on my understanding, refer to "how different the training samples are" and "how many combinations are in each structure". The contribution could be further improved by generalizing the ideas to other mathematical settings, which may introduce other types of composlitionality.
- As pointed out by the author, the equation 5 ignores the potential coupling interactions. How does this correlation affect the generalization error?
- There are some typos in the equations, e.g., in line 215, $|\phi(c_1)-\phi(x_2)|$ should be $|\phi(c_1)-\phi(c_2)|$.

**Questions:**

- Why in figure 4 and 5, M is small (i.e., the maximum examined M is 5), compared with figure 3 (i.e., the minimum examined M is 10)? What happens when M is 10 for GPT-2-XL and Mistral? What will the figure change if increasing the data budget from 4500 to a larger value like 10000?
- I am curious about the impact of different training strategies. If using parameter-efficient tuning instead of fine-tuning, will the effects of the two types of diversity be different?​​
- Why is it necessary to define $s(\cdot)$ as the sub-circuit in $c(\cdot)$? This notation seems not to be used in the rest of this paper.

---

> ### Author Response · Authors · 2025-11-21
>
> We sincerely thank the reviewer for their careful reading and feedback. We will try our best to address these concerns and answer these questions.
>
> 1. Assumption\
> We appreciate the reviewer’s thoughtful concern regarding the geometric assumptions in our theoretical analysis. Our intention is not to claim that real-world compositional functions lie exactly in a d-dimensional Euclidean space, but rather to use this embedding as a standard abstraction that enables tractable generalization bounds. If the covering space is narrow, the theory does not fundamentally change. But a narrow space means easier to cover, i.e., much smaller covering number and structural diversity matters less. We added this assumption in line 215-216 and highlighted in blue.
>
> 2. Generalizing the idea to other settings\
> We appreciate the reviewer's insightful comment and agree that generalizing our framework to other mathematical or structural settings is a promising avenue for future work that could reveal novel forms of compositionality.\
> Our work is intentionally scoped to algebraic circuits to establish a foundational and theoretically tractable framework for the semantic vs. structural diversity trade-off. This clean, controlled setting allows us to isolate core factors and derive tractable bounds. We agree that other domains may introduce new, high-level axes of compositionality. We stated in our revised conclusion (line 504-506, and highlighted in blue) that extending our theoretical framework to incorporate these new diversity dimensions in other mathematical and logical settings is a key goal for future research.
>
> 3. Potential coupling interactions\
> We sincerely thank the reviewer for their sharp observation regarding the omission of potential coupling interactions in Equation 5. This is a critical point for a complete theoretical analysis. The assumption that components are mutually independent (i.e., ignoring coupling interactions) is a necessary simplification for deriving a tractable generalization bound. Our goal was to establish a clear, initial theoretical separation of the fundamental effects of semantic and structural diversity. The current bound serves as a baseline in an idealized setting. We agree that in real-world compositional systems, components are generally correlated or coupled. If a coupling term were included, it would likely be a positive term added to Equation 5, which would generally increase the upper bound of the total generalization error. We expect it to be a decreasing function of both M and N (e.g., $O(1/\sqrt{MN})$, and this will shift the stationary point but not eliminate its existence.
>
> 4. Typos\
> We sincerely thank the reviewer for their careful reading and for catching the typographical errors in the equations. We refined the text (especially section 3.2 line 224, 225, and we highlighted in blue) and we will try our best to ensure all equations are meticulously reviewed and corrected for the final version of the paper.
>
> 5. Experiment Size\
> Figure 3 uses a small-scale encoder-only transformer trained from scratch. This model starts with no prior knowledge, meaning it must learn all fundamental components (the sub-circuits) from the ground up during the training process. To successfully acquire these components and allow for meaningful generalization experiments, the model requires a significantly larger data volume to overcome the initial learning bottleneck of component acquisition. Figure 4 and 5 use Large Language Models (LLMs) (GPT-2-XL, Mistral 7B) that are fine-tuned on the algebraic circuits. These LLMs are pre-trained on massive datasets, giving them a high prior capacity and a substantial advantage in learning the underlying conceptual components (sub-circuits). The paper explicitly states that the component complexity is effectively lower for these larger, more capable models. Because the LLMs already have high capacity and a lower effective $r$, they achieve strong performance and capture the optimal trade-off dynamics with a much smaller data budget. This smaller budget is often chosen to make the fine-tuning process more economical and efficient.
>
> 6. Impact of parameter-efficient tuning\
> We confirm that our experiments on Large Language Models (LLMs) (Figures 4, 5, and 6) already utilize a Parameter-Efficient Fine-Tuning (PEFT) strategy, specifically LoRA, which is the standard practice in this field.
>
> 7. Necessity of $s$
> We think the definition of $s$ as the subcircuit (component) is necessary for the paper's theoretical foundation, even if it does not explicitly appear in the final generalization bounds. The entire Section 3.1 is dedicated to how the model learns these $s$. Additionally, in theorem 3.1, $r$ is defined as the dimension of the subspace where the hypothesis class lives, and it is stated that $r$ characterizes the complexity of the underlying components. $s$ are the explicit mathematical realization of these underlying components that $r$ is meant to capture.

---

> > ### Comment · Reviewer_JbLv · 2025-11-26
> > **thanks for the response**
> >
> > Thank you for your response. Most of my concerns have been addressed.
> >
> > I would like to give more comments:
> >
> > - **response 6 (impact of parameter-efficient tuning):**
> > I found your statements in the revised paper (i.e., Lines 344 and 409) only write "fine-tuning", which is unclear to me.
> > I do not know how you establish your experiments, since different ways of fine-tuning make models behave differently.
> > I understand that the main contributions are the theoretical outcomes. However, you'd better clearly specify the details of the experiments. Without these details, it is difficult to reproduce your results, which hinders your contributions.

---

### Official Review · Reviewer_4mRY · 2025-10-25

**Soundness:** 3
**Presentation:** 3
**Contribution:** 2
**Rating:** 2
**Confidence:** 3

**Summary:**

This paper investigates the role of *compositional generalization* under limited data budgets and introduces a theoretical framework that decomposes data diversity into two orthogonal dimensions: **semantic diversity (M: component variations)** and **structural diversity (N: compositional variations)**. The authors provide generalization bounds for intra- and inter-compositional learning and propose an optimal allocation strategy between M and N. The theoretical claims are validated on synthetic algebraic circuit datasets using both Transformers and large language models (e.g., GPT-2-XL, Mistral 7B), and further extended to real-world reasoning tasks such as GSM8K.

**Strengths:**

The paper is generally well-structured, and the exposition is clear enough to follow the main ideas.

The theoretical formulation provides a reasonable attempt to justify the proposed perspective.

**Weaknesses:**

Several important related works on compositionality and LLM-based structure-sensitive generalization are missing. For example:
	•	“Compositional Semantic Parsing with Large Language Models” (ICLR 2023)
	•	“Does Data Scaling Lead to Visual Compositional Generalization?” (ICML 2025).

The evaluation is limited to GSM8K, which is a relatively shallow reasoning dataset where compositional structure is implicitly assumed rather than explicitly grounded. Thus, it remains unclear whether the observations hold in more rigorous compositional benchmarks (e.g., SCAN, COGS, PCFG-based datasets).


No comparison is made against standard baselines such as common data augmentation techniques or existing data selection strategies. Therefore, it is unclear whether the reported gains originate from the specific “semantic vs. structural balance” or simply from paraphrase-based data expansion.


The experiments are conducted with only two models (“GPT-2-XL (1.5B)” and “Mistral-7B”), which is insufficient to support general claims about scaling trends or the universality of the proposed findings.

**Questions:**

pls see the Weaknesses.

---

> ### Author Response · Authors · 2025-11-21
>
> We sincerely thank the reviewer's constructive feedback. We will try our best to address these concerns.
>
> 1. Missing related work\
> We appreciate the reviewer highlighting these highly relevant, recent works: "[Compositional Semantic Parsing with Large Language Models]" (ICLR 2023) and "[Does Data Scaling Lead to Visual Compositional Generalization?]" (ICML 2025). We added a dedicated discussion of both in the revised Introduction (line 52, 88-89) and Related Work (line 131-135), and highlighted in blue. Specifically, we clarified the following distinction:
> Our work separates semantic vs. structural diversity, providing theoretical bounds, which was not explicitly analyzed in these prior works.
>
> 2. Limited Dataset\
> We appreciate the reviewer's concern regarding the scope of our evaluation datasets. We fully agree that rigorous compositional benchmarks (e.g., SCAN, COGS, PCFG-based datasets) represent a crucial direction for future work.\
> Our justification for the current experimental design is as follows:\
> (1). Utility in Implicit Compositionality: GSM8K was specifically chosen because the compositional structure is implicit (multi-step mathematical reasoning). Our primary objective was to demonstrate the effectiveness of our theoretical framework and data balancing strategy within these implicitly compositional circuits, which are highly relevant to real-world LLM fine-tuning tasks.\
> (2). Existing Rigorous Evidence: To address the need for a rigorous compositional benchmark within the current work, we have already included extensive experiments on algebraic circuits. We argue that these algebraic circuit experiments, which possess a clear, explicit, and rigorous compositional structure, serve the same fundamental purpose as testing on COGS or SCAN: they validate that our theoretical findings hold true even when the structure is not implicit.
>
> 3. Missing Baseline\
> The experiments are precisely designed to test the central hypothesis: the allocation of a fixed data budget. Therefore, comparing different allocations is the correct baseline comparison.\
> (1). Controlled Evaluation: In the real data validation, the GSM8K experiments show that moving the allocation to higher semantic diversity (by pairing a seed with a GPT-4o-generated variant of identical reasoning structure) outperforms a size-matched random subset across all budgets. And existing data selection strategies are fundamentally different. They are utility-driven and select examples based on model-dependent metrics. But our framework proposes structural design principles based on inherent task properties.\
> (2). The Key is the Shift, Not the Expansion: The experiment is a comparison of two equal-sized datasets. It is not about more data (expansion); it is about structurally curating the fixed data budget to prioritize semantic diversity, which the theory predicted would be optimal for the GSM8K task.\
> We added this discussion in the revised text, line 332-336 and highlighted in blue.
>
> 4. Limited Model\
> We appreciate the reviewer's concern regarding the limited number of models used. We respectfully argue that our choice of models was deliberate and strategic, specifically aimed at isolating the effects of our proposed data diversity framework from confounding variables introduced by powerful pre-training.\
> Our primary goal is to validate the theoretical framework and the principled data allocation guidelines, not to benchmark the absolute performance of the latest state-of-the-art models. We specifically selected the GPT-2-XL (1.5B) and Mistral-7B-v0.1 models, which are old generation models. This choice allows us to study the impact of our data design on models that have not been excessively exposed to complex compositional tasks during their massive pre-training phase, which is also observed by reviewer dXnA.\
> We added this discussion in line 407-411 and highlighted in blue.

---

> ### Comment · Reviewer_4mRY · 2025-11-26
> **Official Comment by Reviewer**
>
> Thank you for the rebuttal. The response does not fully address my concerns, so I will keep my original score.

---

### Official Review · Reviewer_dXnA · 2025-11-01

**Soundness:** 3
**Presentation:** 3
**Contribution:** 3
**Rating:** 6
**Confidence:** 4

**Summary:**

This paper studies data diversity for compositional generalization and argues that “more diverse data” is not uniformly better. It formally separates semantic diversity (many surface realizations of the same structure) from structural diversity (many distinct compositional structures), derives generalization bounds for each, and shows that under a fixed data budget K=M ⋅ N, increasing one kind of diversity can hurt if the other is the real bottleneck. The theory (via Rademacher complexity for intra-compositional generalization and covering-number arguments for inter-compositional generalization) yields an error bound with an interior optimum in N, explaining why we need to balance semantic and structural diversity. Experiments on synthetic algebraic circuits and on GSM8K (via GPT-4o–generated variants) confirm that when component complexity is high and compositional complexity is low, semantic diversity wins; when components are simple and recombination is the challenge, structural diversity wins.

**Strengths:**

**1. Intuitive and Robust Theoretical Upperbound of compositional generalization error**

-- Authors study both semantic diversity and structural diversity to disentangle data factors that actually influence compositionality. They established theoretical upper bounds for both intra-compositional and inter-compositional generalization error, which corresponds to the two types of diversity.

-- The overall upper bound of generalization error reveals a crucial fact: increasing diversity without a principled strategy does not necessarily lead to optimal generalization ability, and the training dataset requires a balanced allocation between semantic and structural diversity to achieve data efficiency.

**2. Solid experiment design and strong empirical results**

-- The designed experiments align well with the theory to be tested. The synthetic circuit setup is well controlled and directly tests the theoretical stationary point

-- Real-data validation on GSM8K that shows semantic-augmented subsets can match or beat full-data fine-tuning under the same budget.

**Weaknesses:**

**-- Missing citations:** A few previous works [1,2] also discussed the importance of data diversity for compositional generalization.

[1] Zhou, Xiang, Yichen Jiang, and Mohit Bansal. "Data factors for better compositional generalization." EMNLP 2023.
[2] Akyürek, Ekin, and Jacob Andreas. "LexSym: Compositionality as lexical symmetry." ACL 2023.

**-- Undefined notation of semantic/structure complexity:** See my question below on distinguishing between semantic/structure complexity (d,r) and semantic/structure diversity (M/N).

**Questions:**

-- A few citation format mistakes: For example, authors should use \citep in line 156.

-- When the notations of “semantic/structure complexity (d,r)” are first discussed in Theorem 3.3, they are not properly defined. In Theorem 3.1, it’s not immediately clear why the dimension of $\mathcal{A}$ “characterizes the complexity of the underlying components”. Similarly, in Theorem 3.2, authors mentioned space dimension d characterize structure complexity. Since these two terms are so important for the theoretical contribution of this paper, they should be properly introduced as authors did for semantic/structure diversity (N,M).

-- Most previous work studying compositional generalization trains Transformer networks from scratch rather than from a pretrained checkpoint to eliminate the influence of pretraining data (e.g., a pretrained model may have already learned to do arithmetic compositionally). While Sec 4.1.1 focuses on training encoders from scratch, in Sec 4.1.2, the authors should at least discuss the potential effect of using pretrained models.

-- Can the authors propose a practical estimator of (r,d), even proxy-level, so that the balance between semantic vs structural diversity can be decided automatically for a new task?

---

> ### Author Response · Authors · 2025-11-21
>
> We sincerely thank the reviewer's positive feedbacks, and we will try our best to address these concerns.
>
> 1. Miss citations\
> We have added the suggested references ([Zhou et al., 2023], [Akyürek & Andreas, 2023]) to the related work section and clarified how our approach differs (line 131-135) and highlighted in blue. Specifically, our work separates semantic vs. structural diversity, providing theoretical bounds, which was not explicitly analyzed in these prior works.
>
> 2. Undefined notations\
> We thank the reviewer for pointing this out. We have clarified the notations of semantic/structural complexity (r,d) in Theorems 3.1–3.3:\
> (1). Semantic complexity r measures the number of distinct surface realizations per compositional structure, capturing the internal variability of each component. For example, a three-layer subcircuit allows more variations than a two-layer subcircuit, so components within the three-layer subcircuit have larger r than those in two-layer subcircuits. We added this in line 184-187 and highlighted in blue.\
> (2). Structural complexity d characterizes the diversity of possible combinations of components. For instance, in a three-layer problem, components drawn from two-layer subcircuits have smaller d than components from one-layer subcircuits, reflecting that deeper compositions can generate richer structural combinations. We added this in line 216-218 and highlighted in blue.
>
> 3. Proxy estimator\
> To help practitioners automatically balance semantic vs. structural diversity for a new task, we propose practical proxy estimators:\
> Semantic complexity r: Use the number of layers in the subcircuit as a proxy. We added this in line 184-187 and highlighted in blue.\
> Structural complexity d: Similarly, approximate d using the number of layers in the compositional structure, reflecting the combinatorial richness of possible arrangements. More layers imply more ways to combine components, yielding higher structural complexity. We added this in line 216-218 and highlighted in blue.
>
> 4. Use pre-trained Models\
> We thank the reviewer for pointing this out. In Section 4.1.2, we have added a discussion of the potential effects of using pretrained models (line 407-411, and highlighted in blue). Our experiments in Section 4.1.1 focus on training encoders from scratch to isolate the impact of semantic and structural diversity, and pretrained models may already encode compositional patterns (e.g., arithmetic or reasoning skills). This prior knowledge could reduce the observable effect of additional semantic diversity, since the model has already learned certain component combinations.\
> We emphasize that we do not use state-of-the-art pretrained models because we aim to observe the fine-tuning effects and more clearly isolate the impact of the training data’s semantic and structural diversity. We acknowledge that pretraining effects cannot be perfectly removed, but our setup allows us to study these factors in a controlled manner.

---

> > ### Comment · Reviewer_dXnA · 2025-11-28
> >
> > Thank you for responding to my questions and addressed most of my concerns. I believe this work could benefit from a more formal definition of the structural complexity. I'm inclined to keeping my score.

---

### Official Review · Reviewer_Njx3 · 2025-11-01

**Soundness:** 1
**Presentation:** 3
**Contribution:** 1
**Rating:** 2
**Confidence:** 4

**Summary:**

The work studies compositional generalization in the algebraic circuits context. The work argues for making a distinction between semantic and structural diversity, emhpasizing that semantic diversity by itself may not be sufficient for achieving compositional generalization. An attempt is made to come up with a theoretical framework to include these diversity conditions into a compositional generalization framework.

**Strengths:**

1. The problem of compositional generalization is relevant. The scope (algebraic circuits in controlled settings) is easy to understand and reasonable.
2. The writing is clear

**Weaknesses:**

1. Generalization results (Sections 3.2, 3.3) read like speculation. The analysis essentially extrapolates iid generalization arguments to an __assumed__ configuration within Euclidean space. That is not a genuine distribution-shift generalization proof. I don't see any justification of the assumptions, which makes the results disconnected.
2. Empirical evaluations are tiny, e.g. Fig 3, arguably the graph that should contain most important results, only shows the training progress of three configurations.
3. Related work has made a distinction between diversity and pure scale and its impact on compositional generalization (see, e.g., [1, 2]); these should be discussed and positioned within this work.

[1] Uselis, Arnas et al. “Does Data Scaling Lead to Visual Compositional Generalization?” arXiv:2507.07102.
[2] Zhou, Xiang et al. “Data Factors for Better Compositional Generalization.” arXiv:2311.04420.

**Questions:**

1. Can the analysis be extended to a larger set of variations of of N, M? Current Fig. 3 only shows three configurations.

---

> ### Author Response · Authors · 2025-11-21
>
> We sincerely thank the reviewer for their sharp and constructive feedback. The reviewer raises several excellent points that have helped us significantly improve the clarity and rigor of our work.
> 1. assumption issues\
> Thank you for pointing out the lack of clarity around the assumptions used in Sections 3.2 and 3.3. Our goal is to derive upper bounds under a clearly stated idealized setting that separates intra-compositional (i.i.d. over component variations) and inter-compositional (shift over unseen structures) errors.\
> (1). Justification of the Euclidean embedding assumption.
> In our framework, semantic components correspond to functions in a hypothesis class whose variability is captured by a latent representation phi(c) in R^d. The assumption that components representation in a d-dimensional embedding is standard and does not require components themselves to be Euclidean. We agree this was not clearly stated and we have updated in the revised text (line 215-216) and we highlighted in blue.\
> (2). iid distribution and distribution shift
> Inter-compositional generalization indeed corresponds to a distribution shift (unseen structures). Our analysis does not treat this as i.i.d.; instead, we upper-bound the error using covering numbers over the structure space where the test distribution lies within a bounded set around the training structures. We have added this in the revised text (line 208-211) and highlighted in blue.\
> (3). Connection between the two analyses.\
> The current presentation made the bounds appear “extrapolated” from i.i.d., but the intention is a decomposition:
> i). i.i.d. variance within each structure (semantic diversity), and
> ii). worst-case covering-based shift over structures (structural diversity).
> We added this in the updated text (line 269-273) and highlighted in blue.
>
> 2. Empirical evaluations are tiny\
> We sincerely thank the reviewer for this feedback. We fully agree that expanding the configuration space would significantly strengthen the empirical section. The experiments involve a computationally intensive pipeline of end-to-end data generation, model training, and evaluation for each configuration, making it challenging to provide a complete, exhaustive grid within the short rebuttal window. We will make effort to include and upload any additional experimental results or analysis we can complete during this discussion window.
>
> 3. Missing related work\
> We thank the reviewer for highlighting these important references. In the revised manuscript, we have add a discussion of both Uselis et al. (2025) and Zhou et al. (2023) in the related work section (line 131-135), and we highlighted that in blue. Specifically, we note that these works also investigate the impact of data diversity on compositional generalization. Our work differs in that we formally separate semantic and structural diversity, providing theoretical generalization bounds. This perspective complements prior findings and clarifies when semantic versus structural diversity is more effective, which was not explicitly addressed in the earlier studies.

---

### Meta-Review · Area_Chair_GqoD · 2026-01-04

**Summary:**

The reviewers raised the following concerns:
- Lack justification of Euclidean embedding assumption.
- Limited dataset: Missing experiments on more rigorous compositional benchmarks (e.g., SCAN, COGS, PCFG-based datasets).
- Missing baselines.
- Limited emperical justification: The experiments are conducted with only two models (“GPT-2-XL (1.5B)” and “Mistral-7B”), which is insufficient to support general claims about scaling trends or the universality of the proposed findings.
- Limited experimental diversity: Fig 3, arguably the graph that should contain most important results, only shows the training progress of three configurations. The analysis should be extended to a larger set of variations of of N, M.
- Lack deep analysis and discussion on the distinction between diversity and pure scale and its impact on compositional generalization.
- Missing related work on data diversity for compositional generalization.
- Missing technical details: Undefined notation of semantic/structure complexity; missing experimental details on finetuning.
- Omission of potential coupling interactions in Equation 5.
- Limited scope: it is unclear how to generalize the idea to other settings besides algebraic circuits.

**Reviewer Concerns:**

The following reviewer concerns have been partially addresses by the rebuttal:
- Lack justification of Euclidean embedding assumption: The assumption that components representation in a d-dimensional embedding is standard and does not require components themselves to be Euclidean.
- Missing technical details: defined notation of semantic/structure complexity
- Omission of potential coupling interactions in Equation 5
- Limited scope: future work to generalize the idea to other settings besides algebraic circuits.

The following concerns are still outstanding:
- Limited dataset
- Missing baselines
- Limited emperical justification
- Limited experimental diversity
- Lack deep analysis and discussion on the distinction between diversity and pure scale and its impact on compositional generalization.
- Missing related work on data diversity for compositional generalization.

**Reviewer Scores:**

Reviewer Njx3 would keep the score because among his four concerns, the authors acknowledged two of them, and did not fully addressed the other two regarding limited justification and experiments.

The authors addressed three out of four concerns, and Reviewer dXnA chose to keep his score after the rebuttal.

The rebuttal does not fully address Reviewer 4mRY's concerns, and he keeps his original score.

Reviewer JbLv keeps the score although most of his concerns have been addressed by the rebuttal due to the missing details.

[Note] This paper exceeds 9 page limit and should be desk-rejected.

---

### Decision · Program_Chairs · 2026-01-26

Reject